# Variations in Wood Density, Annual Ring Width and Vessel Properties of *Quercus brantii* Affected by Crown Dieback

**Forough Soheili [1], Stephen Woodward [2], Isaac Almasi [3], Hazandy Abdul-Hamid [4,5] and Hamid Reza Naji [1,*]**

1 Department of Forest Sciences, Ilam University, Ilam 69315-516, Iran; foroughsoheili@gmail.com
2 School of Biological Sciences, University of Aberdeen, Aberdeen AB24 3UU, UK; s.woodward@abdn.ac.uk
3 Department of Statistics, Razi University, Kermanshah 67144-15111, Iran; i.almasi@razi.ac.ir
4 Institute of Tropical Forestry and Forest Products (INTROP), Universiti Putra Malaysia, Serdang 43400, Malaysia; hazandy@gmail.com
5 Faculty of Forestry and Environment, Universiti Putra Malaysia, Serdang 43400, Malaysia
* Correspondence: h.naji@ilam.ac.ir

**Abstract:** Tree decline due to climate change results in physiological weaknesses, attacks by harmful pests and pathogens and threats to forest ecosystem stability. In the work described here, the effects of drought on wood density, tree ring width and variations in vessel morphology are investigated in Persian oak (*Quercus brantii*) in the forest of the Zagros Mountains, Ilam Province, western Iran. Discs are cut from trunks of declined and healthy trees and woodblocks are cut radially from the sapwood near the bark, at a mid-point between the vascular cambium and the pith (middle) and from wood near the pith. Observations are made on transverse sections from the blocks using microscopy. In trees with decline symptoms, wood density is greater than in healthy trees. Furthermore, declining trees have the narrowest ring width, reduced vessel diameter and area and the highest numbers of vessels and tylose in pith towards the bark. It is concluded that changes in anatomical features are associated with the weakening of trees and are components of declining tree health.

**Keywords:** climate change; tree decline; wood density; anatomy; *Quercus brantii*

## 1. Introduction

Forest dieback is a complex phenomenon characterized by decreasing tree growth, defoliation and change in leaf size, shape and color, leading to the immediate or gradual death of trees [1]. Decline and mortality of oak species have been reported for more than a century [2], sometimes occurring in vast areas of forest due to complex and interacting reactions involving environmental stresses, pests and diseases [3]. The Zagros forest in western Iran is the largest oak forest in the world, covering over 5 million ha, dominated by Persian oak (*Quercus brantii* L.). Forest decline has been reported in this oak forest and is ultimately resulting in extensive tree mortality [4]. In the past decade, the impacts of climate change on the Zagros oak forest have led to serious forest disturbance, including dust storms, fires and pest and disease outbreaks, leading to a decline in the main forest tree species of the region [5].

Knowledge of anatomy and function in secondary xylem is a key issue in understanding tree defenses against pathogens [6,7]. Tree ring width records, for example, can provide valuable information about the responses of trees to past environmental events [8,9].

Regulation of vascular cambium activity over the years is crucial to the ability of trees to adjust to fluctuating environmental conditions and, therefore, is highly responsive to climate change. Annual ring structure, however, is stable compared to other morphological characteristics of trees [10,11]. Although variations in tree-ring structure are well understood, little is known of the intrinsic mechanisms responsible for variations [12].

The water potential in the xylem is a key parameter that is widely used to monitor drought stress conditions throughout the plant [13]. The vessel features are influenced

by dryness and drought, as demonstrated previously in oaks [14,15]. Reducing vessel size—and, as a consequence, water conductivity—makes the vessels more vulnerable to cavitation [16]. Anderegg et al. [17] and Giagli et al. [18] demonstrated decreased cambium activity, shortened periods of cambial activity, formation of narrower growth rings and intra-annual wood density fluctuations, resulting from abiotic stresses on the vascular cambium. Abiotic stresses such as dust storms, as consequences of climate change, are also known to affect annual ring widths and vessel diameters and widths in *Q. brantii* [19], possibly due to disruptions to hormonal regulation during wood formation [20].

To prepare evidence-based management plans for the protection of the forest in the Zagros mountains, it is crucial to understand the importance of biotic and abiotic factors and the relationship between the two in forest decline. To our knowledge, there has been no specific study examining this issue. The aim of the work reported here, therefore, is to improve understanding of the impacts of drought stress on the wood structure of *Q. brantii*, focusing on annual growth increment and vessel features. Improving our understanding of how the environment influences host plants and the associated pests and pathogens will help to predict the performance of the forest during drought, to provide practical solutions for forest management.

## 2. Material and Methods

### 2.1. Study Areas

This study was conducted on dead and healthy Persian oak trees (*Quercus brantii* L.) affected by drought stress in three forest regions: Dareh shahr (A), Sheshdar (B) and Chavar (C), all in Ilam Province, Western Iran (Figure 1). The trees had grown as a result of natural regeneration and were sampled from areas with uniform conditions of habitat, altitude and topography, all in the same biosocial classes. The dominant trees in the forest were *Q. brantii*, *Crataegus* sp., *Acer monspessulanum* and a shrub, *Daphne* sp. Important information from the study areas is given in Table 1.

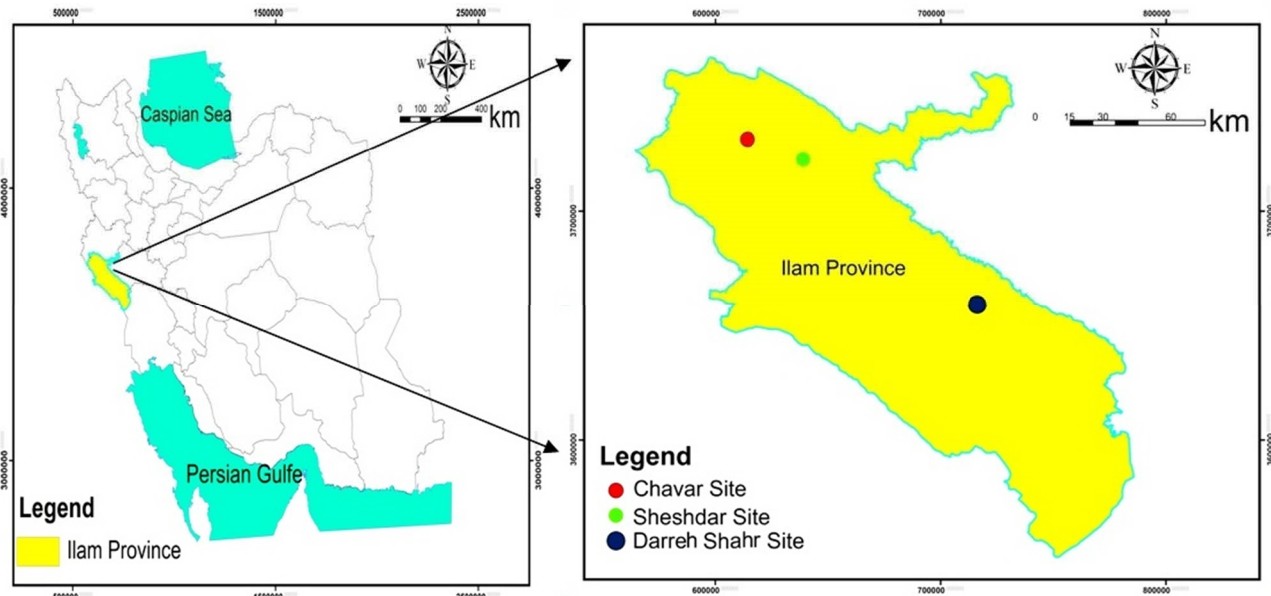

**Figure 1.** Location of the forest sampled in the Ilam Province in western Iran.

**Table 1.** Basic characteristics of the *Quercus brantii* forest sampled for this work.

| Forest Area | Altitude (m) | Longitude | Latitude | Av. Precip. | Av. Temp. | Av. RH |
|---|---|---|---|---|---|---|
| | | | | mm | °C | % |
| Dareh shahr (A) | 933 | 33°3′30″ N | 47°19′30″ E | 465.1 | 19.5 | 41 |
| Sheshdar (B) | 2230 | 33°38′55″ N | 46°30′37″ E | 582.2 | 16.9 | 40 |
| Chavar (C) | 1680 | 33°43′03″ N | 46°14′36″ E | | | |

Av. Precip.: Average precipitation; Av. Temp.: Average temperature; Av. RH: Average relative humidity. Climate data were based on the synoptic meteorological station of Ilam.

### 2.2. Sampling Method

The sampled trees in each area were located close to each other. As the forest is protected, no permission to cut trees in the stands could be obtained. Wood samples, therefore, were collected from the main trunks of trees previously felled by the Department of Natural Resources of Ilam Province under the plan to remove trees showing severe decline from the forest. Eighteen naturally regenerated trees (six trees from each area) of 70–90 years old, with DBH (diameter at breast height) ranging from 30–40 cm were selected. From each sample tree, a disc of approximately five cm in thickness was taken from breast height for further measurements. In the laboratory, each disc was carefully halved transversely, to give two discs for examination of wood density and anatomical properties. For healthy trees, core samples were taken using an increment borer at breast height. Wounds created on the trees were treated with fungicides and pesticides.

### 2.3. Wood Density

From the first part of the disc, blocks with dimensions of $1 \times 1 \times 1$ cm$^3$ were cut continuously (no interval) from the sapwood near the bark, towards the wood near the pith, using a rotary saw. For the samples from healthy trees, the woodblocks were prepared based on the size of the wood cores. Oven-dry wood densities of the samples were calculated based on ISO 3131-1975 (E) [21]. Wood samples were first oven-dried at $103 \pm 2$ °C to reach 0% moisture content and were weighed. Sample blocks were subsequently soaked in distilled water and block volumes were obtained with an accuracy of $\pm 0.01$ g. Prior to immersion, the samples were covered with a thin layer of paraffin wax. The wood density index (WD) was calculated as follows:

$$\rho = \frac{M}{V} = \frac{\text{g}}{\text{cm}^3} \tag{1}$$

where $\rho$, $M$ and $V$ are density, mass and volume, respectively.

### 2.4. Preparation for Macroscopic Measurement

The second part of each disc was prepared using a series of sandpapers (80, 240 and 400) on an orbital sanding machine to attain a smooth, perfectly clear surface [22], before scanning at 2400 dpi resolution. Annual ring widths from 1987–2016 from each disc were measured radially from the pith to the most recent sapwood under a binocular microscope attached to a LINTAB 5 ring-width measuring system (Rinntech Company, Heidelberg, Germany).

### 2.5. Sample Preparation for Sectioning

The disc was cut radially into blocks of 1 cm$^3$ at three positions between the bark and the pith: sapwood near the bark (near-bark), at a mid-point between the vascular cambium and pith (middle) and in the wood immediately adjacent to the pith (near-pith). Each sample block contained approximately 10 growth rings. Blocks were labelled with the area, dieback class and their position on the disc. To better conserve the woodblocks, samples were fixed in FAA solution (50% ethanol: acetic acid: formaldehyde at 1:1:1, *v:v:v*). To

soften the blocks for better sectioning and to improve section quality, samples were boiled at 100 °C for 60 min.

### 2.6. Sectioning

Cross-sections with thicknesses of 20–25 μm were cut using a rotary microtome (POOYAN MK 1110, Iran), transferred to 10% sodium hypochlorite solution for 5 min until totally bleached and then to distilled water until no bleach odor was detectable. Sections were stained in 0.1% ($w/v$) safranin O, washed twice in distilled water for approximately 5 min and dehydrated in an ethanol series of 60%, 85%, 95% and absolute for 15 min each. Dehydrated sections were fixed on slides using Canada balsam [23], prior to observation under an Olympus cx22LED microscope. Photographs of sections were captured using a digital camera (True chrome metrics, Fuzhou, China) attached to a computer. Tangential and radial diameters of earlywood vessels (DEV, μm), earlywood vessels' area (EVA, $\mu m^2$), vessels' density (number of vessels per $mm^2$, NV) and the total number of tyloses (NT, $mm^{-1}$) visible in 30 standard observation fields of 500 $\mu m^2$ were measured from each decline class from each sampling area.

### 2.7. Statistical Analysis

Data were subjected to one-way analysis of variance (ANOVA). The normality and homogeneity of the data were evaluated using Shapiro–Wilk's and Levene's tests, respectively. Independent sample $t$-tests were carried out to determine differences in wood properties among healthy and declined trees at 1% and 5% levels of probability. The data were analyzed in SPSS software, version 21.

## 3. Results

### 3.1. Physical-Anatomical Variables

#### 3.1.1. Wood Density (WD)

There were significant overall differences in WD between healthy and declined trees from the sampled areas ($p < 0.01$; Table 2). The oven-dry WD of declined trees was higher than the WD in healthy trees (Table 3). A decreasing trend in WD from near-pith to near-bark was found in declined trees sampled in all three areas. Comparing healthy trees in areas B and C against declined trees, there was a noticeable reduction in WD in the near-pith vs. middle, with no significant difference in the middle vs. near-bark blocks (Figure 2).

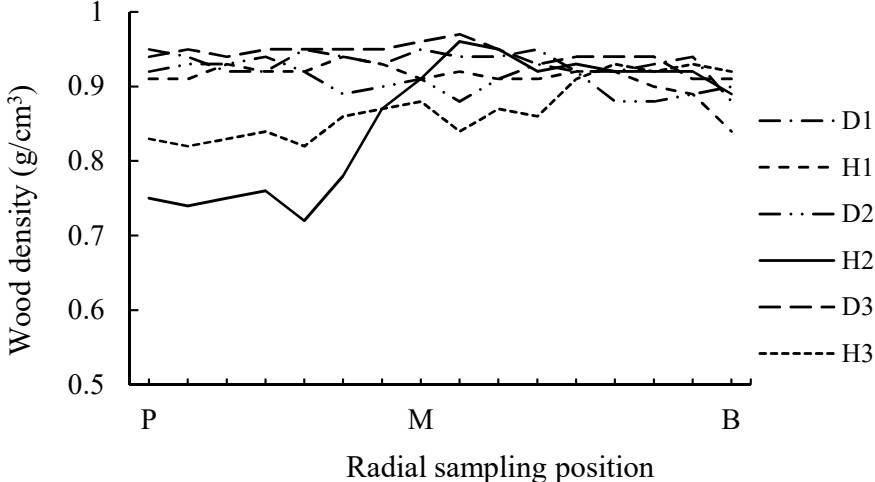

**Figure 2.** Wood density of declined (D1–D3) and healthy trees (H1–H3) of *Quercus brantii* in three areas of the Iranian forest: Dareh shahr, Sheshdar and Chavar. Density was measured in the wood samples from near-pith towards near-bark. H: Healthy trees; D: Declined trees; 1: Area A; 2: Area B; 3: Area C; P: Near-pith; M: Middle; B: Near-bark.

**Table 2.** One-way ANOVA summarizing the significance of wood-anatomical variables (WD: wood density; ARW: annual ring width) between healthy and declined *Quercus brantii* trees affected by drought stress in three oak forest areas in Iran.

| Source of Variation | df | Mean Square | |
| --- | --- | --- | --- |
| | | WD (g/cm$^3$) | ARW (µm) |
| Decline | 1 | 0.57 ** | 115.408 ** |
| Areas | 2 | 0.13 ** | 201.184 ** |
| Decline × Areas | 2 | 0.005 * | 47.531 ** |
| Error | 594 | 0.002 | 1.439 |

WD: wood density; ARW: annual ring width; * Significance at 0.05; ** Significance at 0.01.

**Table 3.** Summary of Student's *t*-test analyses of differences in wood characteristics (means ± SE) between healthy and declined *Quercus brantii* affected by drought stress.

| Attributes | Sampling Group | Mean | Difference | *t*-test |
| --- | --- | --- | --- | --- |
| WD (g/cm$^3$) | Declined | 0.92 ± 0.02 | 0.048 | 5.181 * |
| | Healthy | 0.87 ± 0.06 | | |
| ARW (µm) | Declined | 0.78 ± 0.24 | 1.60 | 5.241 * |
| | Healthy | 2.38 ± 0.54 | | |

WD: wood density; ARW: annual ring width; * $p < 0.01$.

### 3.1.2. Annual Ring Width (ARW)

Comparing ARW over time (Table 2), there was a significant difference ($p < 0.01$) between the means of declined compared with healthy trees across all three areas. The ARWs of declined trees from all sampling areas decreased over time, compared with healthy trees (Table 3). The decreases of ARW in declined trees were very similar in areas A and C. In healthy trees in both areas A and C, there was a similar trend in ARW changes between 2000–2016. Furthermore, in area B, ARWs decreased in declined trees compared to healthy trees (Figure 3).

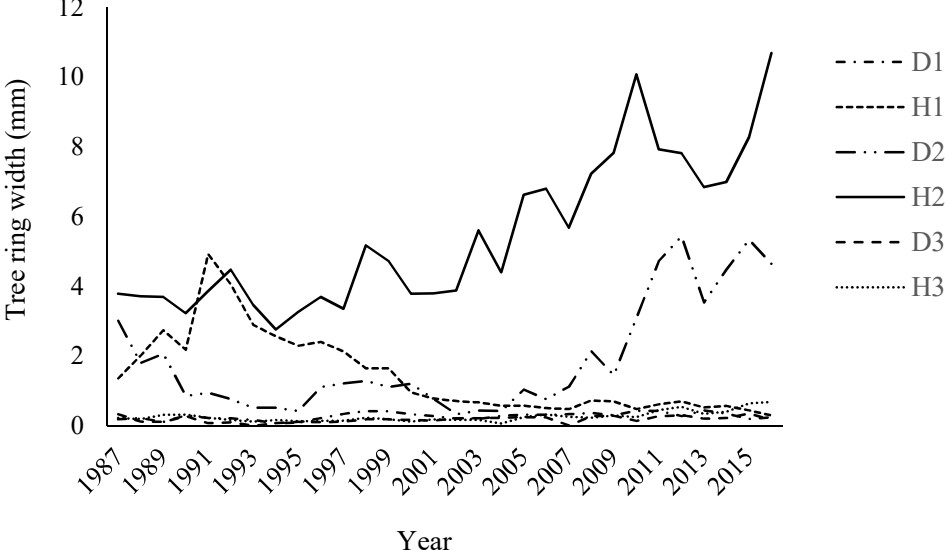

**Figure 3.** Time series of annual ring width of *Quercus brantii* for declined (D1–D3) and healthy trees (H1–H3).

*3.2. Wood Anatomical Features*

3.2.1. Diameter of Earlywood Vessels (DEV)

Variance analysis (Table 4) showed the effect of decline on DEV in the three parts of the stem cross-section: near-pith, middle and near-bark. The sampling area also had a significant impact on DEV in the middle. The interaction effects of decline and area were also significant in the three sampled parts of the trees. The mean DEV of declined trees showed a descending trend from the near-pith to the bark in areas A and C and was constant and similar in healthy trees. In the declined trees from area B, the trend in radial direction was constant and similar, while in healthy trees from area B, there was a decreasing trend from near-pith to near-bark (285.6 down to 262.1 μm, respectively) (Figure 4). In general, the mean DEVs in declined trees from all three sampled parts of the stem were lower than in healthy trees. There were significant differences in DEV between healthy and declined trees in the near-pith, middle and near-bark (Table 5).

**Table 4.** One-way ANOVA of wood-anatomical variables between healthy trees and those with decline at the three study areas.

| The Source of Variation | | df | Mean Square | | | |
|---|---|---|---|---|---|---|
| | | | DEV (μm) | EVA (μm$^{-2}$) | NV (mm$^{-2}$) | NT (mm$^{-2}$) |
| Decline | Pith | 1 | 38.977 ** | 27.906 ** | 1.533 $^{ns}$ | 14.092** |
| | Middle | | 762.045 ** | 127.558 ** | 2.219 $^{ns}$ | 8.477 ** |
| | Bark | | 508.746 ** | 64.223 ** | 11.486 ** | 111.562 ** |
| Area | Pith | 2 | 1.571 $^{ns}$ | 1.71 $^{ns}$ | 3.633 * | 40.897 ** |
| | Middle | | 9.400 ** | 3.984 * | 6.040 ** | 96.150 ** |
| | Bark | | 1.125 $^{ns}$ | 4.442 * | 1.332 $^{ns}$ | 52.205 ** |
| Decline × Area | Pith | 2 | 13.842 ** | 6.941 ** | 0.722 $^{ns}$ | 6.764 ** |
| | Middle | | 32.288 ** | 1.330 $^{ns}$ | 0.367 $^{ns}$ | 2.898 $^{ns}$ |
| | Bark | | 67.281 ** | 4.221 * | 1.964 $^{ns}$ | 10.815 ** |
| Error | Pith | 594 | 12,267.234 | $2.622 \times 10^{11}$ | 2.010 | 0.918 |
| | Middle | | 2808.444 | 2,165,248,861 | 1.730 | 1.387 |
| | Bark | | 3290.675 | 4,289,042,107 | 2.104 | 1.396 |

DEV: Diameter of earlywood vessels; EVA: Earlywood vessels' area; NV: Number of vessels; NT: Number of tyloses; * Significantly different at $p < 0.05$; ** Significantly different at $p < 0.01$; $^{ns}$ No significant difference.

**Table 5.** Differences in wood characteristics (means ± SE) between healthy and declined trees based on Student's *t*-test.

| Anatomical Sampling | | Pith | Middle | Bark |
|---|---|---|---|---|
| **Group Features** | | | | |
| DEV (μm) | Declined | 228.43 ± 15.54 ** | 171.74 ± 6.72 ** | 173.85 ± 7.9 ** |
| | Healthy | 284.88 ± 10.00 | 291.19 ± 6.28 | 282.93 ± 7.18 |
| EVA (μm$^2$) | Declined | 38,754 ± 3391.46 ** | 21,669 ± 2852.86 ** | 22,720 ± 2646.88 ** |
| | Healthy | 74,562.83 ± 13,280.79 | 64,579.56 ± 7088.08 | 65,573.28 ± 10,484.12 |
| NV (mm$^{-2}$) | Declined | 1.62 ± 0.16 $^{ns}$ | 2.09 ± 0.16 $^{ns}$ | 2.24 ± 0.18 ** |
| | Healthy | 1.48 ± 0.14 | 1.93 ± 0.12 | 1.79 ± 0.18 |
| NT (mm$^{-2}$) | Declined | 0.67 ± 0.12 $^{ns}$ | 1.32 ± 0.16 $^{ns}$ | 1.15 ± 0.16 ** |
| | Healthy | 1.02 ± 0.14 | 1.04 ± 0.12 | 0.28 ± 0.06 |

DEV: Diameter of earlywood vessels; EVA: Earlywood vessels' area; NV: Number of vessels; NT: Number of tyloses; ** $p < 0.01$; $^{ns}$ No significant difference.

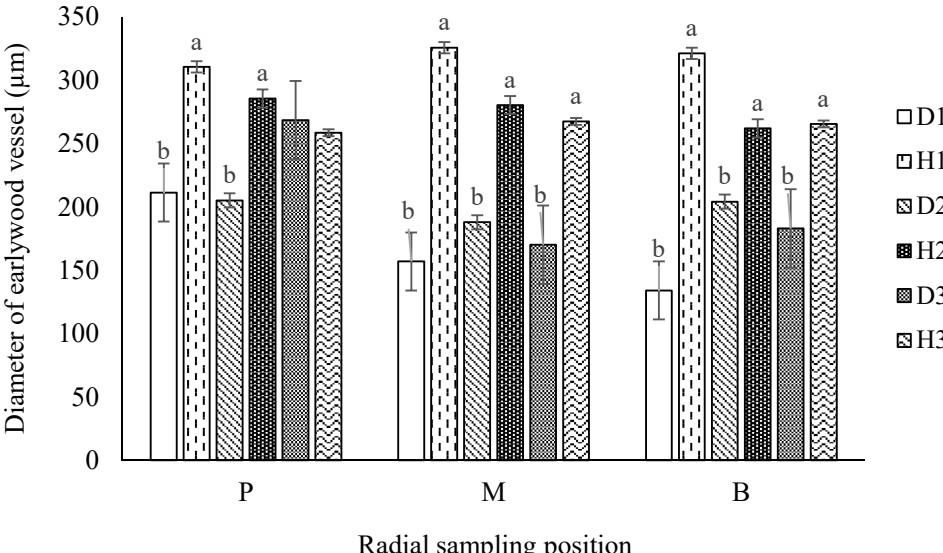

**Figure 4.** Mean diameter of earlywood vessels in declined and healthy trees of *Quercus brantii*. H: Healthy trees; D: Declined trees; 1: Area A; 2: Area B; 3: Area C; P: Near-pith; M: Middle; B: Near-bark. The bar on each column represents the SE (standard error). Columns with no letters on them indicate insignificance.

### 3.2.2. Earlywood Vessels' Area (EVA)

Decline and sampling area had significant effects on EVA in the middle and near-bark, with no significant differences in the near-pith. In addition, interaction effects between decline × area were significant in near-bark samples, but not significant in the middle or near-pith (Table 4). Significant differences were found, however, in the middle samples (Table 4). The mean EVA of declined trees showed a decreasing trend from near-pith to near-bark at all three sampling areas. Healthy trees in areas A and B showed a decreasing trend in EVA from near-pith to near-bark, while an increasing trend was observed in trees from area C (53,816 to 82,628 $\mu m^2$, respectively) (Figure 5). In general, EVA in declined trees from the three sampling positions in stems was lower than in healthy trees. Significant differences in EVA were found between healthy and declined trees in the near pith, middle and near-bark xylem (Table 5).

### 3.2.3. Number of Vessels (NV)

Decline had a significant effect on NV only in near-bark xylem tissues (Table 4). Sampling area, however, had a significant effect on NV in all three sampling positions. There was no significant interaction between decline × area in the near bark, middle and near-pith tissues. The highest NV was in the near-bark in declined trees from all three areas. In the healthy trees, an increasing trend in NV occurred from near-pith to near-bark in samples from area A (1.3 to 1.62 $mm^{-1}$, respectively). In contrast, healthy trees from area B had an increasing trend in NV from near-pith to near-bark (1.39 to 2 $mm^{-2}$, respectively). However, healthy trees in area C showed a fixed trend from near-pith to near-bark (Figure 6). In general, NV in declined trees from the three sampled parts of the xylem tissues (near-pith, middle and near-bark) were higher than in healthy trees. Significant differences in NV were observed between healthy and declined trees only in near-bark xylem (Table 5).

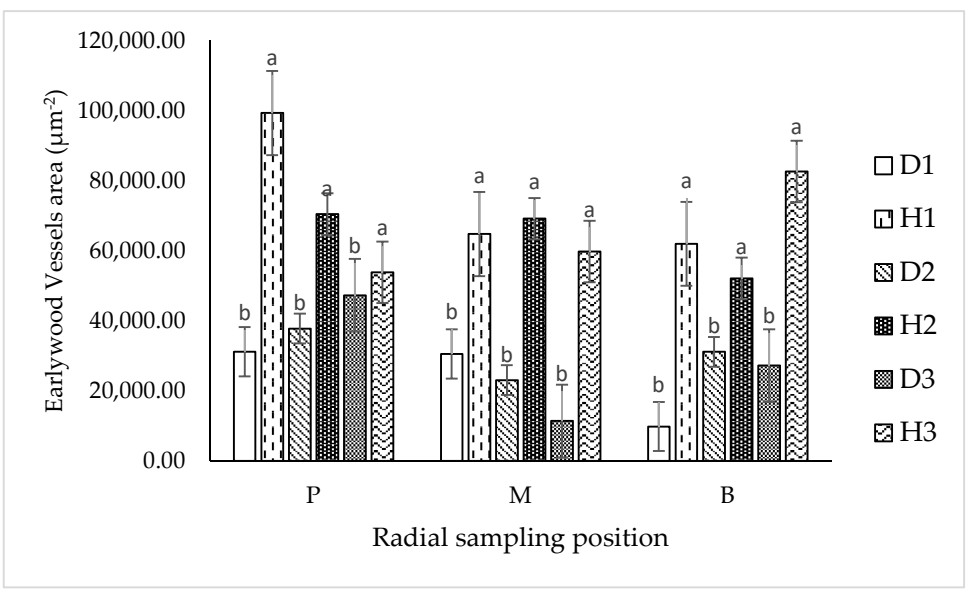

**Figure 5.** Comparisons of mean area of earlywood vessels in declined and healthy trees of *Quercus brantii.* H: Healthy trees; D: Declined trees; 1: Area A; 2: Area B; 3: Area C; P: Near-pith; M: Middle; B: Near-bark. The bar on each column represents the SE (standard error).

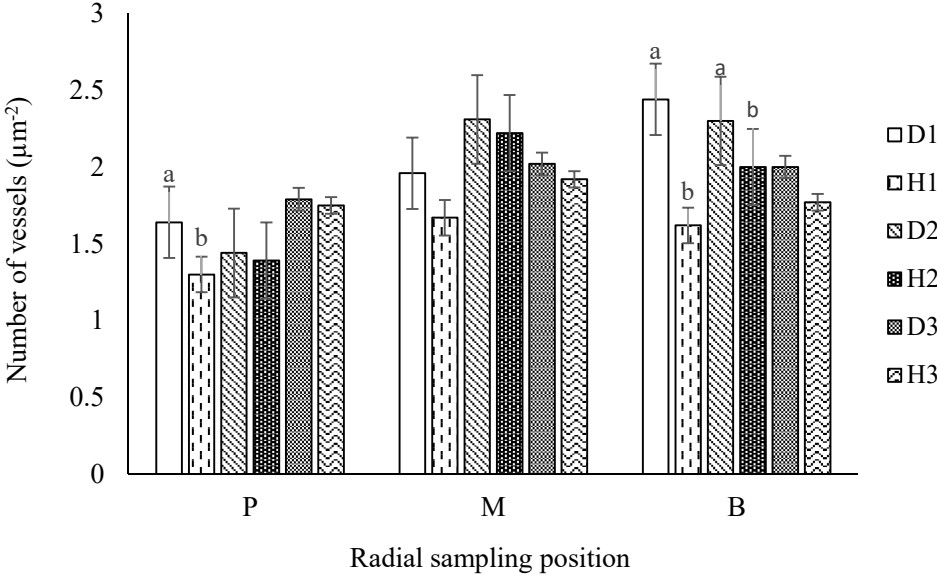

**Figure 6.** Mean number of vessels in declined and healthy trees of *Quercus brantii*. H: Healthy trees; D: Declined trees; 1: Area A; 2: Area B; 3: Area C; P: Near-pith; M: Middle; B: Near-bark. The bar on each column represents the SE (standard error). Columns with no letters on them indicate insignificance.

### 3.2.4. Number of Tyloses (NT)

The effect of decline on the NT was significant in vessels in the near bark, middle and near-pith xylem of stem sections (Table 4). NT varied significantly between the three forest areas in all sampled xylem. Interaction effects, however, was no significantly different only in the middle. In declined trees, NT showed an increasing trend from near-pith to near-bark in trees from all three areas. Healthy trees in areas A and B had the lowest NT in near-bark, increasing towards the pith. Furthermore, NT in healthy trees showed a decreasing trend from near-pith to near-bark in trees from area C (1.8 to 0.65 mm$^{-2}$) (Figures 7 and 8). Thus, higher NTs were found in declined trees from the near-pith, middle and near-bark xylem, compared with healthy trees. Finally, there were significant differences between NT in

healthy and declined trees in near-bark, with no significant differences between the middle and near-pith vessels (Table 5).

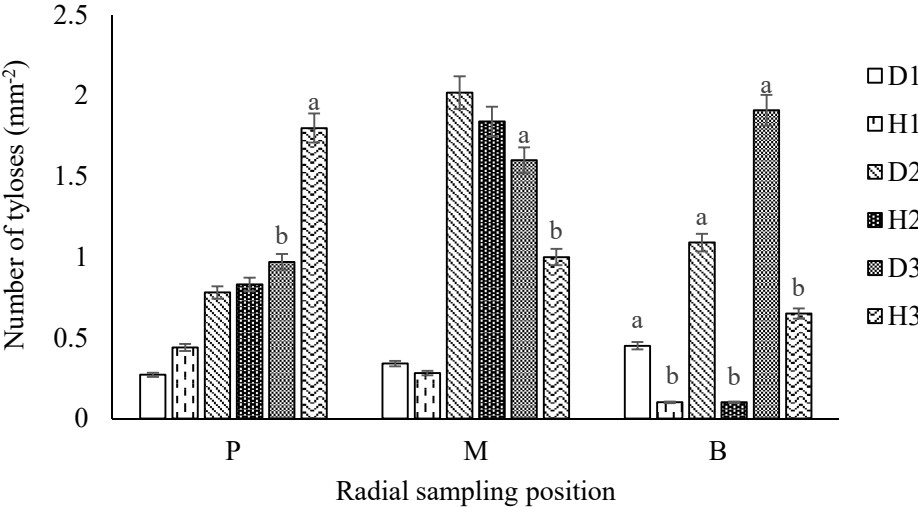

**Figure 7.** Mean number of tyloses in xylem vessels of *Quercus brantii.* H: Healthy trees; D: Declined trees; 1: Area A; 2: Area B; 3: Area C; P: Near-pith; M: Middle; B: Near-bark. The bar on each column represents the SE (standard error). Columns with no letters on them indicate insignificance.

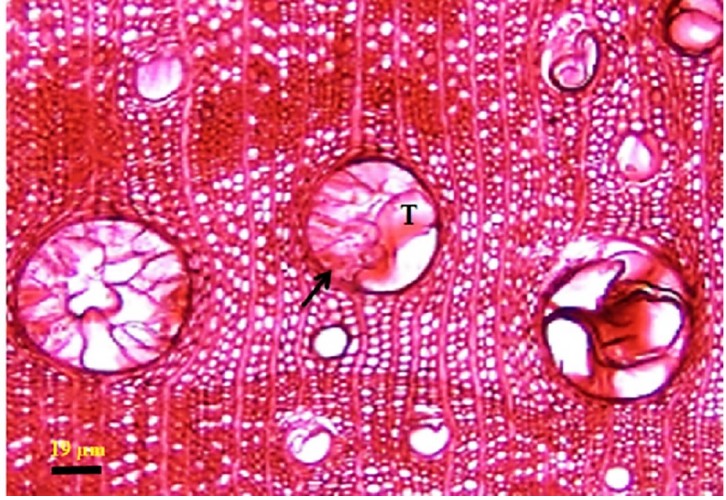

**Figure 8.** Tyloses (T) occluding vessel lumen in transverse section of a declined tree from near-bark tissues.

## 4. Discussion

In this work, variations in annual growth ring and vessel sizes and other wood anatomical features were investigated in *Quercus brantii* affected by drought stress. In general, increases in NV and NT led to increased WD in declined trees, agreeing with previously published work on oak showing that wood density varies as vessel features change [24]. Santiago et al. [25] indicated that an increase in the total surface area of the vessels leads to lower wood density. The occurrence of vascular anomalies as reactions to mechanical or physiological disturbance emphasizes the crucial importance of xylem vessels in tree physiology, as it is thought to play a role in a trading triangle comprising three competing functions: water transport, mechanical support and resistance to embolisms [26].

Density is an important attribute in wood quality and is associated with other determinative properties. Wood density mainly varies through differences in anatomical properties, such as size and distribution of different cells [27], and is strongly related to cell

size and vessel proportions [28,29]. In the work on *Q. brantii* presented here, the DEV and EVA of declined trees from the three sampled forest areas showed a descending trend from near-pith to near-bark. Due to the decrease in the DEV and EVA and increase in the NV in the wood in the near-bark of trees with decline, it appeared that vessel number and density were inversely correlated with average vessel area [30]. It is also known that an increased total surface area of vessels leads to lower wood density [25].

Annual rings are an important source of environmental information [31], enabling variations in climate to be tracked over thousands of years. Narrow rings indicate undesirable environmental conditions (drought or cold periods), whereas wide rings suggest good growth in reasonable conditions (warmer or wetter). In this work, the ARW of trees suffering from decline was significantly smaller than in healthy trees. Although the main driver of this decline in ARW was probably the drought suffered by the trees over a protracted time, it is also possible that changes in the life strategies of endophytic fungi present naturally in the plant tissues, becoming pathogenic under conditions stressful to the host, may further weaken the host plant [32]. The average DEV and EVA in declined trees were smaller than the same features in healthy *Q. brantii*. Decline due to drought could lead to higher NVs in affected trees, as the plants attempt to improve sap flow to the photosynthetic tissues in the crown. A simultaneous reduction of DEV and EVA will lead to a reduction in sap flow in the trees, especially those in decline. Thus, adverse changes in certain wood structural parameters such as vessel diameters are important for hydraulic conductivity. The changes may indicate that the tree genotypes most badly affected are poorly adapted to changing environmental conditions and will, therefore, begin to decline. Furthermore, in response to environmental changes, trees can regulate water conductance further by changing the size and density of the vessels [33,34]. Radial growth of declining *Q. robur* in Poland showed that the average tree ring width and diameter of earlywood vessels in trees that had died was lower than in healthy oaks [35].

Water conductance to leaves in ring-porous trees, such as oaks, is highly dependent on large, functional earlywood vessels. Usually, there is an inverse relationship between the size and the number of vessels, therefore, trees will optimize or increase hydraulic conductivity according to environmental conditions [36]. In trees affected by increasing environmental stresses, the diameter of earlywood vessels and, as a consequence, the hydraulic conductivity, decreases [37]. However, trees may form high numbers of vessels in an attempt to compensate for the problem [38,39]. Likewise, small vessel diameter is highly dependent on climate conditions, including drought [40,41].

Oaks (ring-porous species) usually have large and wide vessels, particularly in the earlywood, and embolism is relatively frequent [42]. An invading pathogen compromising the vascular system leads to the blocking of vessels by the expansion of tyloses from xylem companion cells, sometimes with the secretion of gums and gels to isolate the damaged tissues. The formation of tyloses is an important cause of vascular blockage, preventing sap flow. When the process occurs excessively, it can be a major cause of dieback in the plant. In this study, NT in declined trees from all three forest areas increased in near-bark vessels, compared to xylem in the near-pith. NT was directly related to NV and inversely related to DEV and EVA. Furthermore, there were significant differences in NT and NV in the near-bark of declined trees, but no significant differences in these features in the xylem in the near-pith of trees in decline. It is probable, therefore, that trees that eventually died from decline had the greatest weakness in hydraulic conductivity due to reductions in DEV and EVA and increases in NT and NV. It is well known that anatomical variations caused by abiotic stress decrease with increasing distance from the affected area (here, near-bark towards near-pith) [43].

The presence of tyloses in the earlywood vessels of the annual wood increment influences the reduction of hydraulic conductance in declining oaks [35]. Earlywood vessels contribute to large amounts of sap flow in the main stem and along branches but are vulnerable to embolization [44]. Decreased lumen diameters in severely stressed or infected

oak may assist in controlling water transport and reducing vulnerability to cavitation and embolism [45].

Large-scale and local climate change will have many different effects on individual trees, forest structure and growth performance, including the development and prevalence of fungal diseases [46]. Likely, the increasing occurrence of charcoal disease, caused by *Biscogniauxia mediterranea*, in oaks in the Zagros Mountains is related to recent drought phenomena, emphasizing the need to examine in-depth the relationship between abiotic climatic and biotic factors [7]. Many biotic and abiotic factors are certainly involved in the decline of oak trees, leading Henrique et al. [47] to suggest the complex interacting factors determining the roles of different factors in the survival of this tree.

## 5. Conclusions

Recent studies have not provided comprehensive evidence quantifying the impact of drought stress on Persian oak's anatomical features and wood density. The results presented here, however, have described the effects of abiotic stress on wood density, growth ring width, the diameter of vessels, vessels' area and vessel and tyloses numbers. Notable variation was observed in ARW, DEV, EVA, NT and NV in trees from different sampling areas, in xylem from the near-pith to the near-bark of *Q. brantii* in decline. The WD of declined trees from the near-pith to the near-bark showed a decreasing trend. Water conductivity decreased in response to vessel size and NV increased in declined trees from near-pith to near-bark tissues, contributing to a loss of vitality, and ultimately, the death of trees. The results of this study provide a theoretical basis for future related research on hydraulic parameters in declined trees.

**Author Contributions:** F.S. and H.R.N. carried out the experiment and wrote the manuscript with support from S.W.; I.A. performed the computations and verified the analytical methods; H.A.-H. helped to supervise the project. All authors have read and agreed to the published version of the manuscript.

**Funding:** This research received no external funding.

**Data Availability Statement:** Data Citation: Soheili, F., Woodward, S., Almasi, I., Abdul-Hamid, H., Naji, H.R., 2021; Total data Oak Decline Naji; https://figshare.com/s/4fc5300bf260bbad18bd, accessed on 12 April 2021; doi:10.6084/m9.figshare.14593422.

**Acknowledgments:** We would like to thank A. Hawasi and S. Soltani for their kind help during this work.

**Conflicts of Interest:** The authors declare no conflict of interest.

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
