# Peer review of "Variations in Wood Density, Annual Ring Width and Vessel Properties of Quercus brantii Affected by Crown Dieback"

_forests, doi:10.3390/f12050642_

Round 1
Reviewer 1 Report
The manuscript ‘Variations in Wood Density, Annual Ring Widths and Other Anatomical Properties of Quercus brantii Affected by Crown Dieback’ was submitted for the possible publication in the journal Forests. Submitted research work deals with the impacts of drought stress on annual growth increment and vessel features.
Authors provided interesting data which could be beneficial for the scientific community. However, this manuscript needs revision before acceptance. Below are the comments:
Title: As authors mainly focused on the vessel properties. I suggest you change the title to ‘Variations in Density, Annual Ring Width and Vessel Properties of Quercus brantii Affected by Crown Dieback’.
Abstract: Please use disc only rather mentioning radial disc. Juvenile wood can be transformed to heartwood. Thus, separating juvenile wood, heartwood and sapwood is somewhat vague. Anatomical, physical and mechanical properties vary between juvenile wood and matured wood. Some portion of heartwood could be matured wood. In this case, in that particular height, juvenile wood is also heartwood. Thus, I see a major flaw in this manuscript. Authors need to clarify this issue starting from materials and methods section. One suggestion is to use near pith, middle and near bark rather using confusing term juvenile core wood, heartwood and sapwood.
Introduction: First paragraph needs to have background information about crown dieback as authors investigated dead trees (crown dieback) and healthy trees.
Materials and methods: This part is poorly written and very difficult to follow.
2.2. ‘For healthy trees, a trunk wedge was sampled from the bark and sapwood at DBH’. Not quite clear about this statement. So, samples are not taken upto the pith? In that case, how did your measure those properties in heartwood and juvenile wood of healthy trees?
Did you take core samples? If so, then mention the core diameter?
2.3. Block of ?? cm3. Please be specific and clearly define juvenile core, heartwood and sapwood area according to the comments mentioned in Abstract.
Results: Figure 3: Unit of mm is missing in X-axis. No need of mentioning ‘Density was measured in the xylem of the juvenile core, heartwood, and sapwood’ as you have measured density from pith the bark and showed radial increment in mm.
3.2.1. First line: How did you make sure heartwood is not juvenile wood? Please make this clarification in materials and methods. Use proper terminology and be consistent throughout your manuscript.
Table 5: Differences of wood anatomical properties…DEV: is that radial, tangential or average of both? Make clear about this statement in materials and methods.
Figure 7: Unit of mm is missing in X-axis if you mention X-axis title ‘Distance from pith (mm)’. Need to describe full meaning of D, H and 1-3 as you mentioned in Figure 3.
3.2.2. ‘In addition, interaction effects of decline × site were significant in sapwood, but not significant in the heartwood or juvenile core (Table 4)’. However, Table 4 shows significant differences in the heartwood.
Figure 8. See comments for figure 7.
Instead of vessel density, we typically use Number of vessel per mm2.
Figure 9. Similar comments in figure 7.
Figure 10. Similar comments in figure 7.
The term juvenile core, sap and heartwood must be defined properly in materials and methods and be consistent while describing them throughout the manuscript.
Author Response
Dear editor,
Great appreciation for your knowledgeable comments and suggestions. We did our best to include all issues you noticed. The MS was again edited by a native Eng. editor.
Variations in Wood Density, Annual Ring Width and Vessel Properties of Quercus brantii Affected by Crown Dieback
Journal of Forests;
MS ID: forests-1187634
I hereby respond to the valued reviewers’ comments and suggestions one by one. We thank the reviewers for their knowledgeable comments on our study. We believe that the insightful comments highly improved the quality of our manuscript
Reviewer # 1
Abstract
The requested points were done in the abstract
Introduction
First paragraph needs to have background information about crown dieback as authors investigated dead trees (crown dieback) and healthy trees.
Done
Materials and methods
This part is poorly written and very difficult to follow.
Some major revisions were done.
2.2. ‘For healthy trees, a trunk wedge was sampled from the bark and sapwood at DBH’. Not quite clear about this statement. So, samples are not taken up to the pith? In that case, how did your measure those properties in heartwood and juvenile wood of healthy trees?
Was corrected.
Did you take core samples? If so, then mention the core diameter?
Yes, the descriptions were added.
2.3. Block of ?? cm3. Please be specific and clearly define juvenile core, heartwood and sapwood area according to the comments mentioned in Abstract.
Done
Results
Figure 3: Unit of mm is missing in X-axis. No need of mentioning ‘Density was measured in the xylem of the juvenile core, heartwood, and sapwood’ as you have measured density from pith the bark and showed radial increment in mm.
Done
3.2.1. First line: How did you make sure heartwood is not juvenile wood? Please make this clarification in materials and methods. Use proper terminology and be consistent throughout your manuscript.
We centered our work on the increments from 1987 up to 2016. So, the inner samples may not indicate the heartwood or pith, it is just located in the most inner part in compared to the near bark. The changes were added into the text.
Table 5: Differences of wood anatomical properties…DEV: is that radial, tangential or average of both? Make clear about this statement in materials and methods.
The DEV is measured from the average of two diameters perpendicular to each other. Also, the answer to the second part of the question is specified in Section 2.5.1
Figure 7: Unit of mm is missing in X-axis if you mention X-axis title ‘Distance from pith (mm)’. Need to describe full meaning of D, H and 1-3 as you mentioned in Figure 3.
The unit is not based on the mm, it was considered according to the radial position.
3.2.2. ‘In addition, interaction effects of decline × site were significant in sapwood, but not significant in the heartwood or juvenile core (Table 4)’. However, Table 4 shows significant differences in the heartwood.
Done
Figure 8. See comments for figure 7.
Done
Instead of vessel density, we typically use Number of vessel per mm2.
Done
Figure 9. Similar comments in figure 7.
Done
Figure 10. Similar comments in figure 7.
Done
The term juvenile core, sap and heartwood must be defined properly in materials and methods and be consistent while describing them throughout the manuscript.
Done
Reviewer 2 Report
General remarks
The work described in this paper is interesting, closely linked to ongoing studies on climate change. The study of the reaction of trees to adapt to the changes in progress, especially in the aspects concerning hydraulic conduction, are necessarily carried out at the level of the description, quantification and measurement of the cellular conducting elements.
This work has been well designed and conducted. The most important limitation is in the small amount of trees analysed which makes the interpretation of statistical analyses difficult. On the other hand, the number of anatomical elements measured is instead very high. Probably a better compromise would be to increase the number of trees and concentrate the measurements on sapwood, the area where hydraulic transport occurs in the last years of tree growth.
A formal note that has made reading difficult is the constant change of font, but it does not concern the scientific aspect of the paper.
Detailed notes:
Page 2, top: is climate a pathogen? The sentence suggests such an interpretation.
Page 2, following: the stability of the annual ring structure/anatomy is the basic assumption for anatomical wood identification, why is it a surprise?
Page 3, sampling: why didn’t you choose the core sampling from standing trees?
Page 4, sectioning: the total number of tyloses means the number of vessels occluded by tyloses?
Page 5, figure 3: the whole paper names the sites as A, B, C in the text and 1,2,3 in the captions: is better to be consistent and use the same method.
Figure 6: from the plot it is difficult to see real differences between H and D. Could it be possible that the statistical differences between H and D are due to the different behaviour of H2 that influences all the other H?
Page 7, top: it is not clear the adjective similar to what is related. It is used frequently there, but I couldn’t really reliably relate it to something.
Figure 8: is the only plot (related to EVA) where the differences between H and D are really visible.
Page 9 (VD): “the three tissue types sampled”. What do you mean? Pith-heartwood-bark? I
Wouldn’t call them tissue types (it is always xylem).
Page 9 (NT): the presence of tyloses in sapwood are probably a sign of cavitation, but in heartwood it should be a normal situation due to duraminisation. Isn’t it?
Discussion, beginning: in my opinion and experience wood density is a matter of cell wall percentage (or amount). It is clear that if the vessel are smaller and more frequent there is a higher amount of cell walls. It is also important to underline that wood density in hardwoods is to be connected to fiber: characteristics and amount. But that aspect was not considered in this paper.
Page 11: “Oaks usually have large…”. This is true only for deciduous oaks.
Author Response
Dear Reviewer,
Many thanks for your consideration for fruitful comments and suggestions. We tried to consider in the text all issues you addressed.
Regards, Hamid
Detailed notes:
Page 2, top: is climate a pathogen? The sentence suggests such an interpretation.
The unrelated sentences were omitted and some sentences related to the topic were added. TNX
Page 2, following: the stability of the annual ring structure/anatomy is the basic assumption for anatomical wood identification, why is it a surprise?
Corrected.
Page 3, sampling: why didn’t you choose the core sampling from standing trees?
We chose the sample by core borer.
Page 4, sectioning: the total number of tyloses means the number of vessels occluded by tyloses?
Yes, means the number of vessels occluded by tyloses. We based our measurement on the 30 microscopic field under microscope.
Page 5, figure 3: the whole paper names the sites as A, B, C in the text and 1,2,3 in the captions: is better to be consistent and use the same method.
Done
Figure 6: from the plot it is difficult to see real differences between H and D. Could it be possible that the statistical differences between H and D are due to the different behaviour of H2 that influences all the other H?
Page 7, top: it is not clear the adjective similar to what is related. It is used frequently there, but I couldn’t really reliably relate it to something.
The changes were done.
Figure 8: is the only plot (related to EVA) where the differences between H and D are really visible.
Done
Page 9 (VD): “the three tissue types sampled”. What do you mean? Pith-heartwood-bark? I
Wouldn’t call them tissue types (it is always xylem).
Done
Page 9 (NT): the presence of tyloses in sapwood are probably a sign of cavitation, but in heartwood it should be a normal situation due to duraminisation. Isn’t it?
Yes, you are right.
Discussion, beginning: in my opinion and experience wood density is a matter of cell wall percentage (or amount). It is clear that if the vessel is smaller and more frequent there is a higher amount of cell walls. It is also important to underline that wood density in hardwoods is to be connected to fiber: characteristics and amount. But that aspect was not considered in this paper.
Yes, your comment is fully proper, but we unintentionally used the cell wall features in another paper that is under review.
Page 11: “Oaks usually have large…”. This is true only for deciduous oaks.
The term “ring-porous species” was added.
Reviewer 3 Report
Was the disc obtained directly from the ground or from the breast height diameter?
Is it not better to replace the word "site" with "area", which is more commonly used in forest nomenclature? (Table 1, 2)
Chapter 2.2, why wound was treated by fungicides and pesticides? Whether it's wounds before harvest or after? Will the wounds be in the logs or in the disc?
Chapter 2.3: whether the sample volume was 1 cm3?
Was it not possible to cut samples with a shape whose volume can be measured? You wrote that the volume of the samples was determined by the flotation method. So how much could the wax change the sample volume? The smaller the sample, the greater the impact.
Chapter 2.4: why was the macrostructure determined only for the 1987-2016 interval? So only for 30 years, since the trees were 70-90 years old.
Chapter 2.5: was the plasticization of the samples sufficient without boiling the samples? Oak is a hard species.
Table 2: does it make sense to present the width of the annual rings with an accuracy of 0.001 um? The cell wall thickness in oak is about 2-4 µm.
Fig. 3: the differences between the presented data are almost imperceptible, it is talking about the difference in the types of curves. The disadvantage is also the confusion of declining and healthy in legends.
Are such high density values of completely dry wood correct?
Fig.6: What is the reason for such wide annual rings in wood from the H2 area? The width of the annual increment in ring porous wood decreases with distance from the pith. Similar observations can be made when looking at the width of annual rings below 1 mm. Why such a high density of this wood with such narrow annual rings?
Fig. 7: what is the reason for such a sharp reduction in vessel diameter in H2? In healthy tissue, the diameter of the vessels changes in juvenile wood and in mature wood it is almost the same.
Fig. 9: what is the reason for such a sharp reduction in the number of vessels in healthy tissue H2? With an increase in the width of the annual growth, the number of vessels should not decrease.
Fig. 10: should the number of thyloses in healthy tissue in the sapwood part be close to 0? The formation of thyloses in the wood is associated with the formation of heartwood, their share in sapwood is negligible.
Author Response
Dear Reviewer,
We warmly accepted your comments and suggestions and did our best to consider all of them.
Sincerely, Hamid
Was the disc obtained directly from the ground or from the breast height diameter?
The disc was obtained from DBH. The sentence was corrected.
Is it not better to replace the word "site" with "area", which is more commonly used in forest nomenclature? (Table 1, 2)
Done
Chapter 2.2, why wound was treated by fungicides and pesticides? Whether it's wounds before harvest or after? Will the wounds be in the logs or in the disc?
The reason for using fungicides and pesticides was to prevent the spread of insects and pests into the wood texture after cutting wood. Because, we use increment borer to take out the samples.
Chapter 2.3: whether the sample volume was 1 cm3?
The blocks were taken with dimension of 1×1×1 cm3. It was corrected in the article.
Was it not possible to cut samples with a shape whose volume can be measured? You wrote that the volume of the samples was determined by the flotation method. So how much could the wax change the sample volume? The smaller the sample, the greater the impact.
This is a standard method to measure wood density as it is used in numerous research works reporting in scientific papers.
Chapter 2.4: why was the macrostructure determined only for the 1987-2016 interval? So only for 30 years, since the trees were 70-90 years old.
This research was done in 2016. We meant to determine the effect of drought as a consequence of climate change on wood structure of Persian oaks in widest area of oak forest cross the world. Furthermore, according to the formal reports, the drought was started from almost 2000.
Chapter 2.5: was the plasticization of the samples sufficient without boiling the samples? Oak is a hard species.
The samples had been boiled for about one hour. It was my mistake not to state this statement. TNX
Table 2: does it make sense to present the width of the annual rings with an accuracy of 0.001 um? The cell wall thickness in oak is about 2-4 µm.
The data showed in the Table 2 are not the data measured for annual rings, they are the output of the statistical analyses.
Fig. 3: the differences between the presented data are almost imperceptible, it is talking about the difference in the types of curves. The disadvantage is also the confusion of declining and healthy in legends.
The WD differences between different areas were statistically significant as presented in the Tables 2-3. The data are too close to each other as may not be so clear in the graph.
Are such high density values of completely dry wood correct?
Yes, it is correct.
Fig.6: What is the reason for such wide annual rings in wood from the H2 area? The width of the annual increment in ring porous wood decreases with distance from the pith. Similar observations can be made when looking at the width of annual rings below 1 mm. Why such a high density of this wood with such narrow annual rings?
The features presented for H2 in area B are obviously different from others. This could be due to the site quality. In most features, the results from H2 of area 2 differ from others. In general, during last decade the normal growth of trees in Zagros forest was decreased and influenced because of the drought. The narrow rings in trees could be due to this phenomenon.
Fig. 7: what is the reason for such a sharp reduction in vessel diameter in H2? In healthy tissue, the diameter of the vessels changes in juvenile wood and in mature wood it is almost the same.
I checked the original data. An error happened during the final calculation. It was corrected. Appreciation.
Fig. 9: what is the reason for such a sharp reduction in the number of vessels in healthy tissue H2? With an increase in the width of the annual growth, the number of vessels should not decrease.
I checked the original data, a mistake had occurred during final analysis of the data. The vessel number had to be multiplied by 4 but forgotten. Thanks for your consideration.
Fig. 10: should the number of tyloses in healthy tissue in the sapwood part be close to 0? The formation of tyloses in the wood is associated with the formation of heartwood, their share in sapwood is negligible.
Due to the effect of drought, the probability of occurrence of the tyloses on the sapwood is too high. You are right that the tyloses formation in the sapwood must be close to zero, but please consider that all trees growing in the Zagros forest affecting by drought. So, the probability of their occurrences are high.
Round 2
Reviewer 1 Report
I think the manuscript is improved substantially. However, it needs some clarification and minor changes. Below are my comments:
- Table 1: Is that average RH (%)?
- Sampling method: For healthy trees, Cores samples were taken using.....
- When you have taken core samples from the healthy trees, how did you prepare 1×1×1 cm3 block samples for density measurement? Or you used another sample size? Please provide detail information.
- Figure 3: Y axis unit will be g/cm3.
- Table 4: NV (mm2); EVA (μm2); NT (mm-2)
- Table 5: Correct according to Table 4.
- Figure 9,10: Y axis unit is mm-2.
Author Response
Dear reviewer,
Great appreciation for your careful comments. I did all comments and finalized for further progress. TNX

Reviewer 3 Report
Thank you very much for your answers and taking into account my comments in the text. A lot of good.
Author Response
Thank you very much for the valued reviewer's consideration. I believe that your comments and suggestions helped us to improve the MS. TNX